# Mechanisms Underlying the Rhizosphere-To-Rhizoplane Enrichment of *Cellvibrio* Unveiled by Genome-Centric Metagenomics and Metatranscriptomics

**DOI:** 10.3390/microorganisms8040583

**Published:** 2020-04-17

**Authors:** Yunzeng Zhang, Jin Xu, Entao Wang, Nian Wang

**Affiliations:** 1Joint International Research Laboratory of Agriculture and Agri-Product Safety, the Ministry of Education of China, Yangzhou University, Yangzhou 225009, China; 2Citrus Research and Education Center, Department of Microbiology and Cell Science, IFAS, University of Florida, Lake Alfred, FL 33850, USA; jinxu@ufl.edu; 3Departamento de Microbiología, Escuela Nacional de Ciencias Biológicas, Instituto Politécnico Nacional, C. D. Mexico 11340, Mexico; entaowang@yahoo.com.mx

**Keywords:** *Cellvibrio*, rhizosphere, rhizoplane, metagenome-assembled genome, niche adaptation

## Abstract

Maintaining integrity of the plant cell walls is critical for plant health, however, our previous study showed that *Cellvibrio*, which is recognized by its robust ability to degrade plant cell walls, was enriched from the citrus rhizosphere to the rhizoplane (i.e., the root surface). Here we investigated the mechanisms underlying the rhizosphere-to-rhizoplane enrichment of *Cellvibrio* through genome-centric metagenomics and metatranscriptomics analyses. We recovered a near-complete metagenome-assembled genome representing a potentially novel species of *Cellvibrio*, herein designated Bin79, with genome size of 5.71 Mb across 11 scaffolds. Differential gene expression analysis demonstrated that plant cell wall degradation genes were repressed, whereas genes encoding chitin-degrading enzymes were induced in the rhizoplane compared with the rhizosphere. Enhanced expression of multi-drug efflux genes and iron acquisition- and storage-associated genes in the rhizoplane indicated mechanisms by which Bin79 competes with other microbes. In addition, genes involved in repelling plant immune responses were significantly activated in the rhizoplane. Comparative genomics analyses with five related *Cellvibrio* strains showed the importance of gene gain events for the rhizoplane adaptation of Bin79. Overall, this study characterizes a novel *Cellvibrio* strain and indicates the mechanisms involved in its adaptation to the rhizoplane from meta-omics data without cultivation.

## 1. Introduction

The genus *Cellvibrio* comprises Gram-negative, aerobic, flagellated rod-shaped, and cellulolytic bacteria. Members of this genus have been identified from diverse environments, such as soil, plant root-associated habitats, spring water, and the gut of grass-feeding snails [1,2,3,4,5,6]. At the time of writing, ten recognized species have been described in this genus (based on the List of Prokaryotic names with Standing in Nomenclature; http://www.bacterio.net/index.html), the majority of which were identified from soil environments where plant-derived polysaccharides are abundant. The genus *Cellvibrio*, especially *C. japonicus*, has been intensively studied for its enzymatic potential to degrade a variety of polysaccharides, particularly those present in the plant cell wall, and multiple genes involved in polysaccharide degradation have been identified [1,7].

Mainly owing to root excretion and sloughing, gradients of carbon sources, phytochemicals, as well as other nutrients are formed from the root surface to the surrounding soil, and certain microorganisms are attracted from bulk soil towards roots and form the rhizosphere microbiome [8]. The influences imposed by the plant host on the rhizosphere microbiome are relatively low; however, the root surface, i.e., the rhizoplane, poses different challenges for microorganisms to colonize from the rhizosphere [8]. Plant cells are surrounded by a thick cell wall primarily composed of complex carbohydrates, including cellulose, hemicellulose, lignin, and pectin, and the cell wall plays critical roles in protecting plants against biotic and abiotic stresses, such as pathogen attack and drought [9,10]. Degradation of the plant cell wall is an important strategy for pathogens to infect the plant host [9]. Plants have developed active cell wall integrity-monitoring systems to sense cell wall damage-associated signals, and complex defense responses, including production of reactive oxygen species and activation of hormone-mediated immune responses, will be triggered once the signals are detected [11]. Under such selective forces, only certain microbes from the rhizosphere microbiome can be further enriched and colonize the rhizoplane niche, leading to lower diversity and density of the microbial community in the rhizoplane compared with that of the rhizosphere [8,12]. In our previous study, we surveyed the citrus rhizosphere microbiome at the global scale and observed that *Cellvibrio* was always enriched in the citrus rhizosphere compared with the corresponding bulk soil, regardless of citrus genotype, soil type, and climate, probably resulting from the more abundant plant-driven polysaccharides in the rhizosphere compared with those of the bulk soil [13]. Of note, we observed that *Cellvibrio* exhibited significantly higher relative abundance in the citrus rhizoplane than in the rhizosphere [6].

In this study, we aimed to elucidate the strategies by which *Cellvibrio*, a bacterial genus well known for its plant cell wall degradation ability, is enriched from the rhizosphere to the citrus rhizoplane niche. We performed genome-centric metagenomics and metatranscriptomics analyses using data simultaneously generated from the citrus rhizosphere and rhizoplane samples [6]. An almost complete metagenome-assembled genome (MAG) that represents a potentially novel *Cellvibrio* species was recovered, herein designated Bin79, with estimated genome completeness >99% and contamination rate 0.23%. A large number of plant cell wall degradation-associated genes were identified in the Bin79 genome. Expression profiling of Bin79 in the two niches further revealed the strategies that Bin79 uses to adapt in the rhizoplane.

## 2. Materials and Methods

### 2.1. Metagenome-Assembled Genome Extraction and Curation

In our previous study [6], approximately 120 Gb metagenomic reads (9.25 to 12.04 Gb per sample) from the rhizosphere and rhizoplane samples of six citrus trees were co-assembled and an assembly composed of 17,676,569 contigs with a total length of 10.84 Gb was generated. The assembly and the reads mapping data from individual samples were fed to MetaBAT2, a sequence characteristics and coverage difference-based unsupervised MAG recovery approach, for MAG extraction [14]. A high-quality MAG affiliated with the *Cellvibrio* genus, which was consistently supported by both CheckM and AMPHORA2 taxonomic classification methods [15,16], was obtained and designated *Cellvibrio* sp. Bin79.

The contigs of Bin79 were scaffolded using the Medusa server [17] based on the five publicly available *Cellvibrio* genomes (accessed 2 January, 2017), including *C. japonicus* Ueda107, *C. mixtus* subsp. *mixtus* J3-8, *Cellvibrio* sp. OA-2007, *C. pealriver* PR1 [1,3,4,5], and *Cellvibrio* sp. BR (NCBI Bioproject ID PRJNA81229). The metagenomic reads from sample 2Pm (rhizoplane of tree 2, the sample with the highest abundance of Bin79) were mapped back to the refined scaffolds using Bowtie2 [18], and the alignments were carefully checked using the CLC Genomics Workbench ver. 9 (CLC Bio). If a sequence region was not well supported by the mapped paired-end reads (i.e., the paired-end reads were not concordantly mapped or the average coverage was below five-fold in that region or the coverage in the region was significantly different from the average coverage of the genome), the affected region was discarded. The manually checked sequences were blasted against the NCBI NT database with the cutoff E-value 1e−5, and the scaffolds with no *Cellvibrio* hits were discarded. The reads from 2Pm were aligned to the *Cellvibrio* sp. Bin79 genome using BBMap (ver. 17 May, 2017) [19] with the default parameters and idtag option. The coverage of Bin79 and sequence identity between the mapped reads and genome sequence were determined.

The sequences were deposited in the NCBI database under the Bioproject accession no. PRJNA371449.

### 2.2. Taxonomic Classification

The taxonomic annotation was further determined using the TrueBac ID system, which classified the candidate strain to the species or subspecies level based on whole-genome average nucleotide identity (ANI) and sequence similarity information for two house-keeping genes, *recA* and *rplC*, between the candidate genome and the curated genome database containing 12,444 representative genomes annotated at species and subspecies levels implemented in this system (pipeline ver. 1.92 and db ver. 20190731) [20]. The genome similarity between *Cellvibrio* sp. Bin79 and the five related *Cellvibrio* genomes was determined using the GGDC 2.1 server and fastANI [21,22]. The CGview server was used for the circular representation and comparison of the Bin79 genome against the genomes of *C. japonicus* Ueda107 and *C. mixtus* subsp. *mixtus* J3-8 [23].

### 2.3. Genome Annotation

The genomes of *Cellvibrio* sp. Bin79 and five related *Cellvibrio* strains were annotated using PROKKA with the same parameters [24]. The carbohydrate-active enzyme (CAZyme) gene content was determined for each genome using dbCAN2 [25]. The Clusters of Orthologous Groups of proteins (COG) annotations were assigned to *Cellvibrio* sp. Bin79 genes using the EggNOG server 4.5 [26].

### 2.4. Comparative Genomics Analysis of Cellvibrio sp. Bin79 with Related Cellvibrio Genomes

The pan-genome of the six *Cellvibrio* genomes was calculated using the OrthoMCL method implemented in the GET_HOMOLOGUES software package with parameters identity 50% and coverage 50% [27]. The protein sequences of gene families that contain single-copy genes from each of the six genomes were extracted, then aligned using MUSCLE [28], and the alignments were concatenated using Gblocks 0.91b [29]. A phylogenetic tree was reconstructed based on the concatenated alignment using FastTree2 [30].

The presence and absence matrix of the gene families, together with the phylogenetic tree, were uploaded to the GLOOME server [31], which uses a stochastic mapping method to infer both the total number and the associated probability of gene gains and losses per branch, and the high probability of gains (≥0.8) was analyzed.

Potential regions of horizontal gene transfer (HGT) and genomic islands in the Bin79 genome were identified using Alien_Hunter [32] and the web-based tool IslandViewer [33]. All regions predicted by at least one method were further analyzed.

### 2.5. Differential Gene Expression Analysis of Bin79 Between Rhizosphere and Rhizoplane Niches

The 125 bp paired-end Illumina metatranscriptomic data (7.16 to 9.23 Gb per sample) from the same citrus trees [6] were used for differential gene expression analysis. The metatranscriptomic reads were mapped to the *Cellvibrio* sp. Bin79 genome using Bowtie2 [18], and the count matrix of aligned reads to each of the annotated genes was determined using HTSeq-count [34]. The genes differentially expressed between the rhizosphere and rhizoplane were analyzed using the Poisson distribution method as described by Audic and Claverie [35], and the BH correction procedure to control the false positive rate [36]. The significantly differentially expressed genes between the rhizosphere and rhizoplane (false discovery rate < 0.05 and |log2(fold change)| ≥ 1) were further analyzed.

## 3. Results and Discussion

### 3.1. Recovery and Curation of the Almost Complete Metagenome-Assembled Genome Bin79

In our previous study [6], we observed that several genera, including *Cellvibrio*, showed significantly higher relative abundance in the rhizoplane microbiome, a niche characterized by more intimate host–microbe interactions, compared with the rhizosphere microbiome, and the mechanisms underlying their rhizosphere-to-rhizoplane enrichment were further revealed at the whole microbial community level. Recent developments on MAG recovery and analysis methods enabled us to perform genome-centric analyses on the structure and function of the microbiome, which can provide more detailed and accurate information to reveal the mechanisms underlying niche adaptation of microbes compared with whole community-based analyses [37,38,39]. To gain a better understanding of the strategies used by the adapted microbes for the citrus rhizosphere-to-rhizoplane enrichment, we sought to recover MAGs from the metagenome data and compare the gene expression profiles of the MAGs between the rhizosphere and rhizoplane niches. Using MetaBAT2 [14], an unsupervised MAG extraction tool, we recovered a high-quality MAG affiliated with the *Cellvibrio* genus, supported by both CheckM and AMPHORA2 results [15,16]. *Cellvibrio* is a crucial member of the citrus rhizosphere microbiome, as revealed by our global-scale citrus microbiome analysis, and is further enriched to the rhizoplane niche [6,13]. Therefore, we further refined this *Cellvibrio* MAG to assemble a high-quality genome [40] to investigate the strategies used for rhizoplane niche adaptation.

The initial *Cellvibrio* MAG was composed of 41 contigs with a total length of 5717,923 bp; after reference-assisted scaffolding and manual curation, the curated genome contained 11 scaffolds with a total length of 5713,429 bp, with N50 of 1.80 Mb, and was designated *Cellvibrio* sp. Bin79 (Figure 1 and Table 1). The 16S rRNA gene was not recovered for Bin79, owing to the complexity of the soil microbial community and known issues with metagenome assembly and binning of 16S rRNA gene sequences [41]. The genome assembly was confidently supported by the reads mapping result (Appendix A). The estimated completeness and contamination rate of the Bin79 genome were 99.14% and 0.23%, respectively, as determined by CheckM [16] based on the 507 Gammaproteobacteria-specific marker gene set integrated in this tool.

### 3.2. Phylogenetic Analysis of Bin79

Bin79 was confidently assigned to the *Cellvibrio* genus based on the whole-genome sequence-based classification system TrueBac ID and two marker gene-based taxonomic classification methods [15,16,20]. However, no species-level annotation could be assigned to Bin79 by the TrueBac ID system based on the currently available 12,444 species and subspecies annotated genomes implemented in this database; the digital DNA–DNA hybridization (dDDH) and genome-wide ANI results demonstrated that Bin79 displayed low genomic similarities with the five *Cellvibrio* genomes (highest similarity with *C. mixtus* subsp. *mixtus* J3-8, dDDH value 24.6% (<70%) and ANI value 82.84% (<95%)) (Appendix A) [20,21,22], indicating that Bin79 represents a potentially novel species of *Cellvibrio*.

The genome-wide BLAST results also demonstrated that Bin79 harbored multiple specific genomic regions that were not present in the related strains *C. mixtus* subsp. *mixtus* J3-8 and *C. japonicus* Ueda107. Many of these Bin79-specific genomic regions were probably acquired via HGT as suggested by Alien_Hunter and Islandviewer analyses (Figure 1) [32,33].

### 3.3. Gene Expression Comparison of Bin79 between the Rhizosphere and Rhizoplane Niches

*Cellvibrio* is known for its enzymatic potential to degrade polysaccharides, including those present in the plant cell wall, mainly as a result of the activity of glycoside hydrolases (GHs) [1,5,7]. Two hundred and forty-four GH-encoding genes, such as members of the plant cell wall degradation enzyme GH3 family, were identified in the Bin79 genome, which is comparable to the number identified in the five related *Cellvibrio* genomes (Appendix A). These Bin79-related strains, such as J3-8 and Ueda107, were originally isolated from environments rich in plant-derived materials, e.g., the gut of grass-feeding snails, and field soils. Additional experiments demonstrated that these strains are capable of degrading plant cell wall-associated polysaccharides [7,42].

Interestingly, we observed that Bin79 was more abundant in the citrus rhizoplane than the rhizosphere (paired *t*-test, *p* < 0.01) (Figure 2). The abundance difference of microbes across citrus trees was also observed in our previous global scale citrus rhizosphere microbiome study [13]. The integrity of the plant cell wall is actively monitored by the plant host, and cell wall damage triggers complex defense responses to protect the host against attacks by cell wall-degrading pathogens [11]. To determine how Bin79, a microbe with plant cell wall-degradation capability, is enriched from the rhizosphere to the citrus rhizoplane, gene expression profiles of Bin 79 in the rhizosphere and rhizoplane samples were determined and compared based on the metatranscriptomic data. In total, 119 rhizoplane upregulated and 384 rhizoplane downregulated genes were identified (Appendix A).

Multiple plant cell wall degradation-associated genes, such as those encoding enzymes degrading cellulose (CAZy name Cbp2D), arabinan (Gly43C, Gly43D, Gly43E, Gly43G, and Gly43N), and xylan (Xyn10D, Xyn11B, Gla67A, and Abf51A) [1,7], exhibited significantly higher expression levels in the rhizosphere compared with the rhizoplane (Figure 3). Furthermore, genes encoding the type II secretion system (T2SS) (*gspDCFGHIKLME*), which is the sole system that is responsible for exporting the plant cell wall-degrading enzymes out of the cell [7,43], were more actively expressed in the rhizosphere (Figure 3). Previous research demonstrated that the amount of secreted cellulase is much less when simple carbon resources, such as glucose and starch, are present compared with the presence of complex polysaccharides, such as cellulose [43,44]. Notably, numerous sucrose and glucose transport and utilization-associated genes, such as *gntP*, *gntT*, and *glu13A*, were upregulated in the rhizoplane, whereas the expression levels of genes involved in T2SS synthesis were significantly lower than those in the rhizosphere. Two genes that are critical for chitin degradation in *Cellvibrio*, namely LPMO10A and Chi18D [45] belonging to chitinases, exhibited significantly higher expression levels in the rhizoplane than in the rhizosphere (Figure 3). Chitinases cover different kinds of hydrolytic enzymes that break down the glycosidic bonds in chitin. Chitin is an essential structural component of the cell wall and plays a critical protective role in fungi [46]. It is probable that plants specifically select for *Cellvibrio* to inhibit infection by fungal pathogens. A recent study demonstrated that chitin degradation is a critical function of the disease-suppressive microbiome for suppression of fungal pathogens [47]. Chitinolytic bacteria are important resources for screening biocontrol agents and have great potential to work as sustainable fungicides in the field [48]. In addition, several genes involved in indole-3-acetic acid (IAA) synthesis, including the key genes *trpC* and *ipdC*, were identified in the Bin79 genome, suggesting that Bin79 possibly benefited the plant host by producing IAA [49]. However, IAA production did not contribute to the rhizosphere-to-rhizoplane enrichment process of Bin79, since the expression of IAA synthesis genes did not show difference between rhizoplane and rhizosphere.

Besides the observed differential expression of genes involved in carbohydrate metabolism between the rhizoplane and rhizosphere, the sulfate transporter-encoding gene (*cysPUWA*) and genes involved in assimilatory sulfate reduction (*cysNDCHJI* and *sir*) exhibited higher expression levels in the rhizosphere (Figure 3). Sulfur is an essential element for biosynthesis of cellular constituents in bacteria, for instance, it is a universal substrate for cysteine biosynthesis. Furthermore, with the genes that were exhibited, significantly higher expression levels in the rhizosphere were found to be overrepresented in the COG category “Amino acid transport and metabolism” (Figure 4), and several genes involved in amino acid synthesis, such as *thrC* (threonine synthesis), *dapD* and *dapC* (lysine), *carA* (arginine), and *ilvD* (isoleucine, leucine, and valine), were observed to exhibit higher expression levels in the rhizosphere (Figure 3). In line with these observations, more genes belonging to the function category “Translation, ribosomal structure, and biogenesis” showed higher expression levels in the rhizosphere, and several ribosome assembly-associated genes, such as *rpsK*, *rpsQ*, *rplT*, and *rplN*, showed higher expression levels in the rhizosphere (Figure 3 and Figure 4). These results suggested that the cells in the rhizosphere had to synthesize more nutrients by themselves compared with those in the rhizoplane, and further supported the notion that the microbes colonizing the rhizoplane enjoy superior access to plant root exudates, which are rich in easy-to-use nutrients for the microbes, compared with those microbes colonizing the rhizosphere [6,8].

Gene enrichment analysis demonstrated that more genes that were significantly upregulated in the rhizoplane were assigned to the functional category “Cell motility” (Figure 4). Multiple genes involved in flagellar assembly, including *flgAMFGHIK* and *fliCEFH*, were upregulated in the rhizoplane; *flrB* and *flrC*, which encode a two-component signal-transducing system that regulates flagellar assembly [50], exhibited higher expression levels in the rhizoplane (Figure 3). Importantly, the expression levels of *atoS* and *atoC* were increased in the rhizoplane compared with those in the rhizosphere. The AtoSC signal-transducing system is critical for flagellar synthesis and chemotaxis, and is essential for bacteria to efficiently migrate towards high concentrations of attractants, such as plant-derived simple carbon sources and amino acids, and away from repellents [51]. In contrast, more genes belonging to the category “Cell wall/membrane/envelope biogenesis” showed higher expression levels in the rhizosphere, and several genes involved in lipopolysaccharide (LPS) synthesis and export, including *lpxD*, *lpxL*, *lpxB*, *kdsA*, and *lptB*, were observed to be more actively expressed in the rhizosphere (Figure 3 and Figure 4). Lipopolysaccharide is crucial for bacteria to form biofilms in the environment [52]. These results suggest that Bin79 actively moves in the rhizoplane via chemotaxis and the flagellar system to access the plant-derived easy-to-use nutrients, whereas it forms a biofilm in the rhizosphere.

The rhizoplane, however, poses different challenges for bacteria to colonize from the rhizosphere niche. On one hand, the plant immune response exerts stronger effects in the rhizoplane than in the rhizosphere; on the other hand, the richer nutrients in the rhizoplane attract microbes, leading to fierce competitive interactions among the microbes [8]. Common strategies used by microbes against other competitors include limiting resources such as iron and producing antimicrobial compounds [12]. Interestingly, the functional category “Defense mechanisms” was overrepresented in the gene group that showed higher expression level in the rhizoplane (Figure 4), indicating that Bin79 encountered stronger selective forces in the rhizoplane. CirA, a critical receptor for the ferri-siderophore complex [53], exhibited higher expression level in the rhizoplane compared with the rhizosphere. The only bacterioferritin (Bfr)-encoding gene identified in the Bin79 genome, PROKKA_00691, was more strongly induced in the rhizoplane compared with the rhizosphere (Figure 3). Bfr is activated by iron and can store iron at concentrations much higher than the feasible solubility limits of iron (III) in bacterial cells [54]. Sequestration of iron in the immediate vicinity represses the growth of competitive microbes, thus benefiting the iron-storing bacteria in the competitive microenvironment [12]. Multiple genes involved in multidrug resistance and efflux pumps, such as *nepI*, *emrAB*, *acrAB*, and *tolC*, showed higher expression levels in the rhizoplane (Figure 3). The energy-dependent efflux pumps are extremely important for bacteria to survive in the presence of antibiotics and toxic compounds [55]. AcrAB–TolC is the main multidrug efflux machinery in *Escherichia coli* to endow the bacteria with resistance to a broad spectrum of antibiotics [56].

An organic hydroperoxides (OHPs) resistance enzyme-encoding gene *ohr*, and its regulator-encoding gene *ohrR*, exhibited higher expression levels in the rhizoplane. Production of OHPs is an important component of plant defenses and OHPs are highly toxic to bacteria [57]. The expression of *ohr* is specifically induced by OHPs and the gene product is critical for bacterial survival in the presence of OHPs [58]. Interestingly, the *ohr* and *ohrR* genes were observed to be uniquely present in the Bin79 genome but were not identified in the five related *Cellvibrio* genomes (Figure 5A). BLAST results demonstrated that both genes are homologous to their counterparts in *Rheinheimera* (Blastp, identity 72% for *ohr* (WP_127019447.1) and 70% for *ohrR* (WP_127022368.1)). *Rheinheimera* is frequently isolated from plant-associated materials including rhizosphere soil and roots [59,60], indicating that the *ohr* and *ohrR* genes of Bin79 might have been acquired from other soil-borne microbes, such as *Rheinheimera*, via HGT, which endowed Bin79 with resistance to plant OHPs. The results of Alien_Hunter analysis further demonstrated that the *ohr* and *ohrR* genes were located within a genomic region that was acquired via HGT.

### 3.4. Influence of HGT on the Citrus Rhizoplane Adaptation Capability of Bin79

The evolution of the bacterial accessory genome is believed to be dominated by HGT, and HGT is an important evolutionary force for bacteria to adapt to the environment [61,62]. To investigate how HGT has contributed to Bin79 adaptation to the citrus rhizoplane, we compared the Bin79 genome with those of the five related *Cellvibrio* strains. The core genome of the six *Cellvibrio* accessions comprised 2018 orthologous groups, whereas the pan-genome contained 9210 orthologous clusters. The pan-genome matrix was uploaded to GLOOME [31] to identify the gene gain and loss events (Figure 5B). Extensive gene gain events were observed in the Bin79 clade that were absent in the J3-8 clade, originally isolated from snail guts. The difference in gene gain events between Bin79 and J3-8 probably results from the differential selection pressure in the rhizoplane and the snail gut. It is expected that the environment in the snail gut is relatively stable compared with soil. A large proportion of the Bin79 horizontally acquired genes (746 out of 1184) were further identified to be distributed as acquired genomic islands/clusters via HGT as predicted by both Alien_Hunter and IslandViewer [32,33]. In addition to the *ohr* and *ohrR* genes, several horizontally acquired genes that play important roles in stress resistance and environment adaptation were identified by both phylogenomic-based analysis (GLOOME) and sequence characteristics-based analysis (Alien_Hunter/IslandViewer). For example, PROKKA_04149, a cadherin-like gene (Pfam accession number PF16184) that is important for cell–cell adhesion [63], exhibited a higher expression level in the rhizoplane than in the rhizosphere. In addition, the *traG* gene, which is essential for bacteria to exchange DNA content by conjugation [64], was identified as a Bin79-specific horizontally acquired gene and exhibited a significantly higher expression level in the rhizoplane than in the rhizosphere, indicative of contribution of gene exchange events to the rhizoplane adaptation of Bin79.

## 4. Conclusions

Overall, in situ gene expression profiling analysis of Bin79, a near-complete genome belonging to the *Cellvibrio* genus recovered from the citrus root-associated microbiome, revealed the strategies adopted by *Cellvibrio* to colonize the rhizoplane. The strategies include gene expression upregulation or gene gain for optimal utilization of simple carbon sources, repelling plant immune responses, and competition from other microbes. Based on the information indicated from the genome sequence, customized culture media were designed and strain isolation experiments were performed but without success despite multiple attempts. Currently, the majority of microbes in the soil microbiome remain uncultivated. However, the recovered genomes from metagenomics analyses have deepened our understanding of the diversity and function of the uncultured microorganisms [65]. This study provides such a useful example to characterize a bacterial strain from metagenomic and metatranscriptomic data without culturing and to understand the mechanisms by which it adapts to the rhizoplane.

## Figures and Tables

**Figure 1 microorganisms-08-00583-f001:**
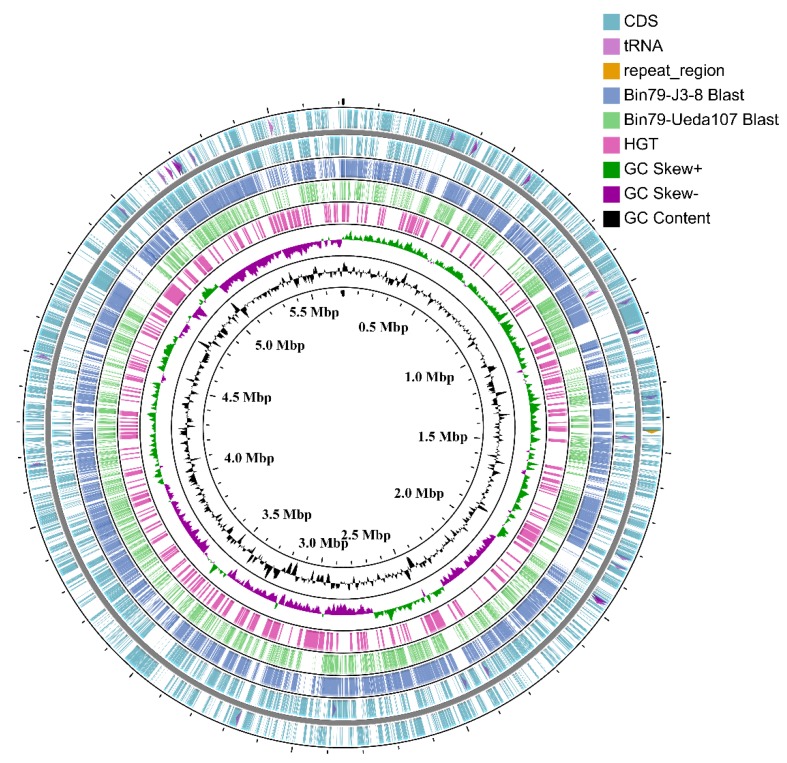
Circular representation of the *Cellvibrio* sp. Bin79 genome. The contents of the feature rings (starting with the outermost ring) are as follows: Ring 1 and Ring 2, features from the forward and reverse strands, respectively, CDSs are drawn as arcs, and tRNAs and repeat regions are drawn as arrows; Ring 3 (*Cellvibrio mixtus* subsp. *mixtus* J3-8) and Ring 4 (*Cellvibrio japonicus* Ueda107) show BLAST comparison results (BLASTN, E-value 1e−20) with the Bin79 genome, the BLAST comparison results are drawn with partial opacity, and darker regions indicate that multiple hits to the corresponding sequence region were observed; Ring 5 shows the genomic regions acquired via horizontal gene transfer; Ring 6 shows GC skew; and Ring 7 shows GC content. The plot was drawn using the CGview server [23].

**Figure 2 microorganisms-08-00583-f002:**
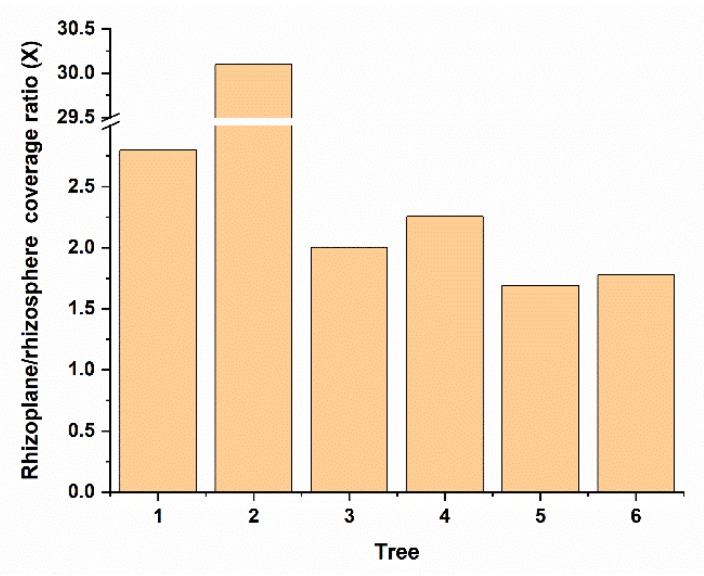
Abundance comparison between rhizosphere and rhizoplane niches of Bin79. The genome coverage ratios between the rhizosphere and rhizoplane niches are calculated by dividing the number of reads mapped from a rhizoplane sample by the number of reads mapped from the corresponding rhizosphere sample.

**Figure 3 microorganisms-08-00583-f003:**
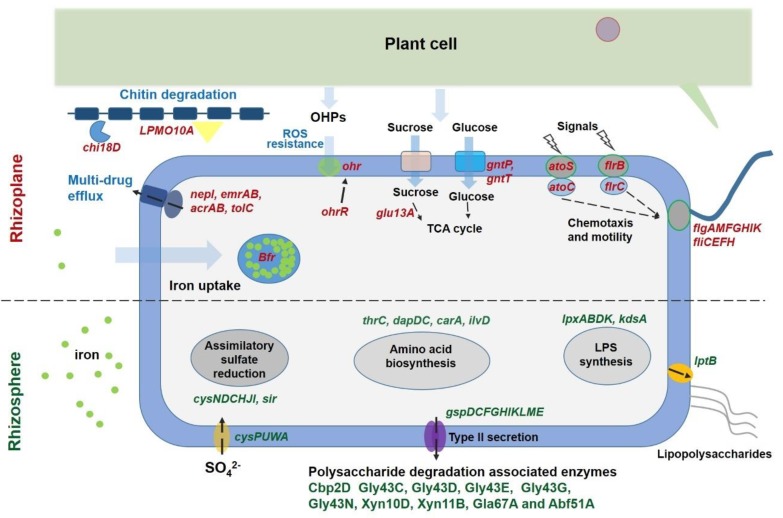
Overview of the genes with expression level differing between rhizoplane and rhizosphere habitats. Genes with higher expression level in the rhizoplane niche are red colored, while genes with higher expression level in the rhizosphere habitat are dark green colored. Iron is indicated by green circles, chitin is indicated by blue line-linked rectangles. The arrows denote influx and efflux of compounds.

**Figure 4 microorganisms-08-00583-f004:**
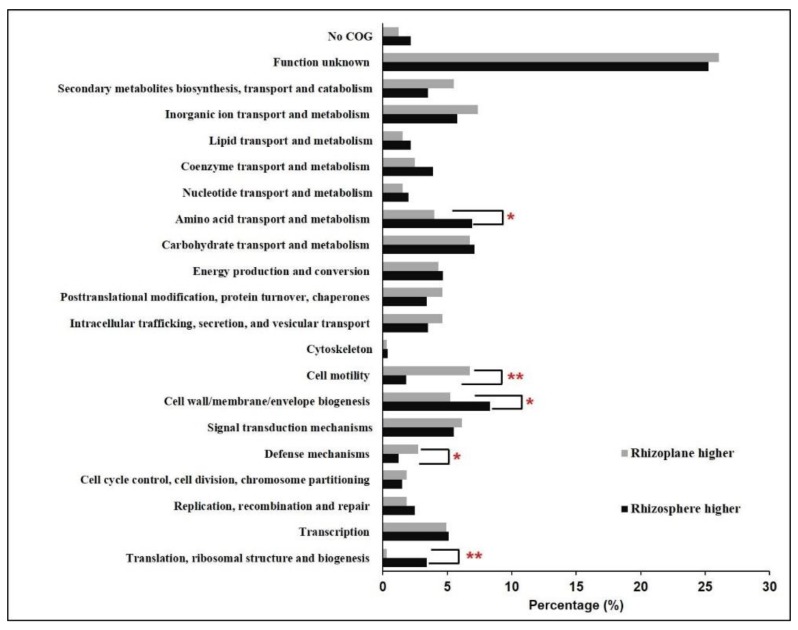
Functional distribution of the *Cellvibrio* sp. Bin79 differentially expressed genes between the rhizoplane and rhizosphere. *, *p* < 0.05; **, *p* < 0.01, determined by one-tailed Fisher’s exact test.

**Figure 5 microorganisms-08-00583-f005:**
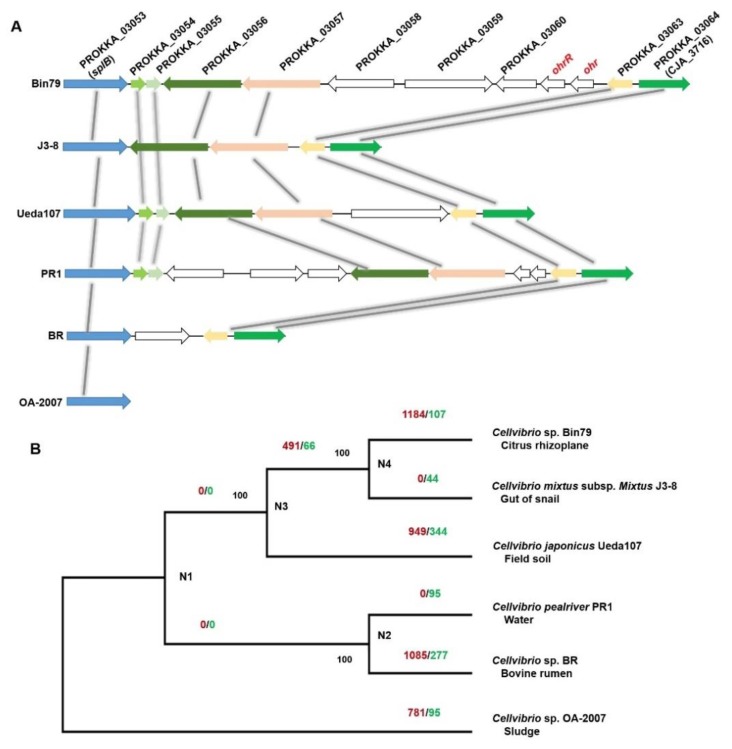
(**A**) Comparison of the *ohr* and *ohrR* gene clusters of Bin79 with related *Cellvibrio* genomes. Conserved and homologous genes (>50% identity) are colored and linked. (**B**) Results of gene gain/loss analysis. Numbers on the branches indicate the number of gene gain (left, red colored) and loss (right, green colored) events at the divergence event. The maximum likelihood phylogenetic tree was reconstructed based on the concatenated sequences of 2018 core genes. Bootstrap support values from 1000 replicates are indicated at the nodes. The isolation sources of the strains are indicated under the strain name.

**Table 1 microorganisms-08-00583-t001:** Features of the *Cellvibrio* sp. Bin79 genome.

Features	Value
Total reading base pairs (bp)	5713,429
Scaffold number	11
N50 (bp)	1795,691
Read coverage	15.68
GC content (%)	46.21
tRNAs	36
Protein-coding sequences	4754

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
