# Peer review of "Mechanisms Underlying the Rhizosphere-To-Rhizoplane Enrichment of Cellvibrio Unveiled by Genome-Centric Metagenomics and Metatranscriptomics"

_microorganisms, 2020, doi:10.3390/microorganisms8040583_

Round 1
Reviewer 1 Report
Nice job on the manuscript. Work was presented logically. Approach was sound, data are interesting. You reported on an interesting aspect of how bacteria can survive in close proximity to the root using HGT. I would have liked you to expand a bit on the importance of the chitanase aspect and the potential for some sort of mutualistic benefit between the plant and bacteria. Paper was well written.
Author Response
Response: Thanks for this valuable comment. We add more information on the importance of Chitinases. Several genes involved in IAA production were identified in the Bin79 genome, suggesting that Bin79 possibly benefited the plant host by producing IAA. However, IAA production did not contribute to the rhizosphere-to-rhizoplane enrichment process of Bin79, since the expression of IAA synthesis genes did not show difference between rhizoplane rhizosphere.
The sentences have been changed to “Two genes that are critical for chitin degradation in Cellvibrio, namely LPMO10A and Chi18D [45] belonging to chitinases, exhibited significantly higher expression in the rhizoplane than in the rhizosphere (Figure 3). Chitinases cover different kind of hydrolytic enzymes that break down the glycosidic bonds in chitin. Chitin is an essential structural component of the cell wall and plays a critical protective role in fungi [46]. It is probable that plants specifically select for Cellvibrio to inhibit infection by fungal pathogens. A recent study demonstrated that chitin degradation is a critical function of the disease-suppressive microbiome for suppression of fungal pathogens [47]. Chitinolytic bacteria are important resources for screening biocontrol agents and have great potential to work as sustainable fungicides in the field [48]. In addition, several genes, involved in indole-3-acetic acid (IAA) synthesis, including the key genes trpC and ipdC, were identified in the Bin79 genome, suggesting that Bin79 possibly benefited the plant host by producing IAA [49]. However, IAA production did not contribute to the rhizosphere-to-rhizoplane enrichment process of Bin79, since the expression of IAA synthesis genes did not show difference between rhizoplane and rhizosphere.”
Reviewer 2 Report
The paper “Mechanisms underlying the rhizosphere-to-rhizoplane enrichment of Cellvibrio unveiled by genome-centric metagenomics and metatranscriptomics” by Zhang et al represents an interesting contribution towards understanding the complex interactions occurring in rhizosphere and rhizoplane of citrus plants. The paper is well written and technically sound. It’s a pity that 16S rRNA sequences could not be deconvoluted. Comments are detailed below.
Page 1. Abstract. Lines 25-26.
Please reformulate: ”In addition, genes involved in resistance to plant immune responses were significantly activated in the rhizoplane.”
It’s kind of odd to say involved in resistance to a response. Who initiated the interaction? Likely the bacteria, which has the means to prevail over, repel, fend off plant immune responses.
Page 7 Figure 3.
“Polysaccharide derogation associated enzymes”.
Why “derogation”? Maybe “degradation” would be a more suitable word.
Page 7 lines 245-249.
“Furthermore, the genes that were exhibited significantly higher expression activities in the rhizosphere were found to be overrepresented in the COG category “Amino acid transport and metabolism” (Figure 4). In addition, several genes involved in amino acid synthesis, such as thrC (threonine synthesis), dapD and dapC (lysine), carA (arginine), and ilvD (isoleucine, leucine, and valine), were observed to exhibit higher expression levels in the rhizosphere (Figure 3).”
Please remove “in addition” and reformulate the two sentences. Amino acid metabolism comprises anabolic (amino acid synthesis) and catabolic pathways (amino acid degradation). Therefore, amino acid synthesis is part of amino acid metabolism. On cannot say “In addition”.
Page 7, lines 250-254.
“In line with these observations, more genes belonging to the function category “Translation, ribosomal structure, and biogenesis” showed higher expression activities in the rhizosphere, and several ribosome assembly-associated genes, such as rpsK, rpsQ, rplT, and rplN, showed higher expression activity in the rhizosphere (Figures 3 and 4), which was indicative of a greater translational burden for the cells colonizing the rhizosphere.”
If amino acid synthesis is upregulated, it is clear that protein synthesis (translation) should be also upregulated. These processes are associated with cell growth and division in presence of sufficient nutrients. I cannot see how this can be a “translational burden”. It means that bacteria are thriving; energetically, amino acid synthesis is a costly process that requires a lot of resources! Please reformulate.
Page 7, lines 255-256
The results further support our notion that the microbes colonizing the rhizoplane enjoy superior access to plant root exudates, which are rich in easy-to-use nutrients for the microbes, compared with those microbes colonizing the rhizosphere [6].
I would say that this is a general concept rather than “our notion”. I would suggest to use “the notion” and add other citations (maybe a review).
On page 7, lines 246, 252 and 253, and page 9, lines 292 and 298 I suggest using “higher expression” and not “higher expression activities”.
Page 10, line 345-346
Remove “and” and add a comma after “responses”.
… simple carbon sources, resistance to plant immune responses, and competition from other microbes.
Author Response
Thanks for your valuable comments.Attached please find our replies to the comments.

Reviewer 3 Report
The manuscript titled "Mechanisms underlying the rhizosphere-to-rhizoplane enrichment of Cellvibrio unveiled by genome-centric metagenomics and metatranscriptomics" resents valuable and unique results, it is well-written and can be published without changes.
The manuscript titled "Mechanisms underlying the rhizosphere-to-rhizoplane enrichment of Cellvibrio unveiled by genome-centric metagenomics and metatranscriptomics" is devoted to study of the mechanisms underlying the trends in rhizosphere-to-rhizoplane modification of Cellvibrio genome and expressed genes. Near-complete metagenome-assembled genomes representing a potentially novel species of Cellvibrio were compared by differential gene expression analysis. Comparative genomics analyses with five related Cellvibrio strains showed the importance of gene gain events for the rhizoplane adaptation.
Revival of the genus Cellvibrio (Winogradsky 1929) gen. nov., with an amended description proposed by by Blackall, L , Hayward, A.C. & Sly, L.I. (Journal of Applied Bacteriology 1985.59, 81–97) attracted attention to the bacteria that degrade the plant cell wall and synthesize an extensive portfolio of hydrolytic enzymes that display highly complex molecular architectures. Sequence analysis of related saprophytic soil bacterium Cellvibrio japonicus reveals 130 predicted glycoside hydrolases that target the major structural and storage plant polysaccharides.
Cellvibrio species are considered as a potential bacteria for one‐step bioconversion of hemicellulose polymers to value‐added products. This is particularly interesting as most industrially relevant bacteria are neither able to depolymerize wood polymers nor metabolize hemicellulose. Adaptation of those bacteria to rhizoplane show the trend for possible further evolution of the Cellvibrio as plant-associated or even plant pathogenic microorganism.
Cellvibrio is a rare object for genetic and genomic studies (I counted about 10 articles published within 2016-2020 and devoted especially to this genus), and any additional information will have high value.
Author Response
Thanks for your encouragement. We very appreciate it!